# Characteristics of a Particle's Incipient Motion from a Rough Wall in Shear Flow of Herschel–Bulkley Fluid

Alexander Seryakov [1], Yaroslav Ignatenko [2,*] and Oleg B. Bocharov [3]

1 OFS Technologies, 1st Krasnogvardeisky pr., 22, 123112 Moscow, Russia; rednex@yandex.ru
2 Baker Hughes, Baker-Hughes-Straße 1, 29221 Celle, Germany
3 IWEP SB RAS, Molodezhnaya str. 1, 656038 Barnaul, Russia; bocholeb@gmail.com
* Correspondence: yaroslav.ignatenko@gmail.com; Tel.: +49-152-314-2-36-46

**Abstract:** A numerical simulation of the Herschel–Bulkley laminar steady state shear flow around a stationary particle located on a sedimentation layer was carried out. The surface of the sedimentation layer was formed by hemispheres of the same radius as the particle. The drag force, lift force, and torque values were obtained in the following ranges: shear Reynolds numbers for a particle $Re_{SH} = 2$–200, corresponding to laminar flow; power law index $n = 0.6$–1.0; and Bingham number $Bn = 0$–10. A significant difference in the forces and torque acting on a particle in shear flow in comparison to the case of a smooth wall is shown. It is shown that the drag coefficient is on average 6% higher compared to a smooth wall for a Newtonian fluid but decreases with the increase in non-Newtonian properties. At the edge values of $n = 0.6$ and $Bn = 10$, the drag is on average 25% lower compared to the smooth wall. For a Newtonian fluid, the lift coefficient is on average 30% higher compared to a smooth wall. It also decreases with the increase in non-Newtonian properties of the fluid, but at the edge values of $n = 0.6$ and $Bn = 10$, it is on average only 3% lower compared to the smooth wall. Approximation functions for the drag, lift force, and torque coefficient are constructed. A reduction in the drag force and lifting force leads to an increase in critical stresses (Shields number) on the wall on average by 10% for incipient motion (rolling) and by 12% for particle detachment from the sedimentation bed.

**Keywords:** particle; sedimentation; rough surface; shear flow; Herschel–Bulkley fluid; drag force; lift force; Shields number

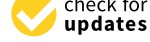



## 1. Introduction

The problem of optimal flow regime determination for liquids carrying a solid component often arises and is extremely important in medicine, the food industry, river hydraulics, and in petroleum engineering. The force characteristics of a single particle in the flow are used as a rule to describe the behavior of an ensemble of particles in a moving fluid. Of particular interest is the beginning of the movement of a grain located on the sedimentation layer. Incipient motion from the surface should start when the local flow velocity exceeds some critical value. An alternative method for determining the beginning of the particle motion is the Shields number, which characterizes the local shear stress on the wall. The parameters of the incipient motion are necessary, for example, in the oil and gas industry when drilling wells to effectively describe the cuttings carried out by the drilling fluid. The problem of determining forces and torque for a spherical particle in a fluid flow has been known for a long time in fluid dynamics. More than 150 years ago, Stokes [1] derived a formula for the drag force acting on a sphere in a plane-parallel unrestricted flow. Almost a century later, Rubinow [2], in addition to the Stokes force, presented expressions for the lift and torque acting on a spinning sphere moving in a viscous fluid. For the shear flow, Saffman [3] determined the value of the lift force for a spherical particle. The corrected expression for calculating the Saffman force is given in [4].

Since the flow of suspensions occurs, as a rule, in bounded channels, an important subject is the influence of the wall on the particle's behavior. Thus, Goldman et al. [5] derived asymptotic solutions of the Stokes equations for the case of a rotating particle moving parallel to a smooth wall in a shear flow. The authors used lubrication theory methods to estimate the forces acting on the sphere with small Reynolds numbers and distances to the surface.

McLaughlin [4] and Cherukat [6] derived the velocity of transverse migration of a particle in a shear flow of Newtonian fluid bounded by one and two walls. Comparison with experimental data on lift force measurements showed satisfactory agreement.

Krishnan and Leighton in their paper [7] generalized the mathematical calculations of Cherukat and McLaughlin to the case where the particle touches the wall. The authors obtained six integral coefficients for determining the lifting force by considering rotation, movement of the particle parallel to the wall, and shear flow of the surrounding fluid. The study showed that the particle is expected to be located at a stable distance above the wall in the channel due to the reduction in the lift force produced by the rotation.

Zeng et al. [8] presented a systematic numerical simulation of hydrodynamics around a stationary particle over a smooth wall in a shear flow of Newtonian fluid and results for a particle moving parallel to the wall. The calculations were performed for particle shear Reynolds numbers from 2 to 250, at which the flow has a laminar character. The authors obtained correlations for the drag and lift force coefficients.

In a continuation of the studies by Zeng et al., Ignatenko et al. [9] extended the simulation for a stationary, moving, and rotating particle on a smooth wall to the case of a Herschel–Bulkley shear fluid flow. In the variant where the sphere is stationary, the correlations of the drag and lift force coefficients in the range of Reynolds numbers from 2 to 200 were developed. These correlations agree with the formulas of Zeng et al. for a Newtonian fluid [8].

When a particle moves over the sediment bed, it rolls from one cavity to another over the other particles. At the same time, there is a rotation around the contact point of the underlying particles. The positivity of the tilting or rollover torque of the forces relative to the contact points is the condition for the movement initiation. If we refer to the detachment of the particle, then the lifting force combined with the Archimedean force must be greater than the force of gravity. Thus, Clark and Bickham, in [10], derive an expression for the critical local velocity near the particle; once this is exceeded, the detachment occurs. An alternative criterion of detachment is determining the Shields number for a particle [11] and comparing it with the critical values. In [12], the critical Shields numbers corresponding to rolling and detachment were calculated based on the force characteristics of a particle located on a smooth wall in a Herschel–Bulkley shear fluid flow. In the real situation, when cuttings start to move, the particles are located on a sedimentation layer that consists of the same particles, so their force characteristics may differ significantly due to the roughness of the surface. Lee and Balachandar [13] presented the results of modeling of the shear Newtonian fluid flow around a sphere located on a substrate of hemispheres of the same diameter. It was found that the drag force coefficient increased up to 10% when approaching a rough surface compared to the case of a smooth wall. The authors noted a decrease in the lifting force in the shear flow with small Reynolds numbers, of the order of two, as the distance from the sphere to the surface decreased. It should be noted that work both supporting such experimental observations [14] and in disagreement with them [15,16] can be found in the literature.

In a continuation of the studies of particles moving over a rough surface in a Newtonian fluid flow, Balachandar et al. [17] simulated a turbulent flow for the shear Reynolds numbers $Re_\tau$ of the order of 180. It should be emphasized that for the particle size relative to the computational domain in [13], the shear Reynolds number $Re_{SH}$ has been found to be 661. The roughness was described using an extended (in comparison to that presented in [13]) hemispheres pattern with a diameter equal to the particle. Direct numerical simula-

tion of the turbulent flow around the sphere on the rough surface was used to establish the fact that the lift force significantly increases due to the sweep events.

In [18], a simulation of the flow of an ellipsoidal particle in a shear flow of Newtonian fluid over a surface of hemispheres was performed. The coefficients of the drag and lift forces and the torque as a function of the shear Reynolds number, ellipsoid orientation angle, and distance from the surface were determined. The authors constructed approximation formulas for the forces and torque that depend on the main flow parameters and geometric characteristics.

The works described above are mainly devoted to the flow of a particle in Newtonian fluid flow, both on smooth and on rough surfaces. This study is a continuation of the work of [13], related to the flow over a sphere on a rough surface in a Newtonian fluid shear flow, and the work of [9], related to the flow over a particle on a smooth surface in a Herschel–Bulkley fluid shear flow. In the present work, we study the forces and the torque acting on a sphere located on a rough surface in laminar shear flow of a Herschel–Bulkley fluid. The rough surface is modeled as in [13] as hemispheres located in nodes of hexagonal honeycombs. The fluid parameters are consistent with the characteristic parameters of drilling fluid flow in well bores. The shear particle Reynolds numbers $Re_{SH}$ vary within the range 2–200, which corresponds to the laminar flow regime. The study of the influence of a rough surface and rheology on the drag force, lifting force, and torque forces and the development of corresponding approximation functions are the main goals of the current work. Analyses of the criteria of incipient motion (rolling) and detachment of a particle from a rough surface are also carried out.

This article is structured as follows. The article starts with the Introduction in Section 1, followed by the Problem Statement and Numerical Algorithm and Computational Mesh in Sections 2 and 3. These are followed by the Results and Discussion in Section 4, which is divided into subsections devoted to the discussion of Forces and Torque (Section 4.1), their Approximation (Section 4.2), and an Analysis of Incipient Motion and Lift-Off Conditions (Section 4.3). Section 5 provides the Conclusions. A list of the main notation is given in the Nomenclature at the end.

## 2. Problem Statement

The steady state laminar shear flow of a Herschel–Bulkley fluid over a rigid sphere with diameter $d$ placed over a rough wall is considered. The sphere is fixed in space. The rough surface is formed by hemispheres of equal diameter $d$ arranged in a honeycomb sequence; see Figure 1. The incoming flow has a constant shear rate $G$ directed along the x axis with zero velocity at the x–y surface and hemispheres. The fluid viscosity depends on the local shear rate and is described by the Herschel–Bulkley rheological model $\mu = (k\dot{\gamma}^n + \tau_y)/\dot{\gamma}$, where $k$ is the flow consistency factor, $n$ is the power law index, $\tau_y$ is the yield stress, and $\dot{\gamma}$ is the second invariant of the strain rate tensor $\boldsymbol{S} = 0.5(\nabla\boldsymbol{u} + \nabla\boldsymbol{u}^T)$. The laminar steady state flow of a viscous incompressible fluid is described by the Navier–Stokes and continuity equations, which can be written in dimensionless form:

$$\begin{cases} (\boldsymbol{u} \cdot \nabla)\boldsymbol{u} = -\nabla p + \dfrac{1}{Re}\nabla(2\mu\boldsymbol{S}); \\ \nabla \cdot \boldsymbol{u} = 0, \end{cases} \tag{1}$$

where $\boldsymbol{u}$ is a dimensionless velocity vector; $p$ is dimensionless pressure, and $Re = \rho u_{ch} L_{ch}/\mu_{ch}$, $u_{ch}$, $L_{ch}$, and $\mu_{ch}$ are the characteristic velocity, length scale, and viscosity, respectively. Taking $L_{ch} = d$, $u_{ch} = Gd$ and $\mu_{ch} = \mu(\dot{\gamma} = G)$, we obtain the particle shear Reynolds number $Re = Re_{SH} \equiv \dfrac{\rho \cdot Gd \cdot d}{(kG^n + \tau_y)/G}$ that can be transformed to $Re_{SH} = \dfrac{\rho G^{2-n} d^2}{k(1 + Bn)}$; here, $Bn = \dfrac{\tau_y}{kG^n}$ is the Bingham number. Thus, the problem can be characterized by three dimensionless parameters that are varied in the ranges $Re_{SH} = 2$–200, $n = 0.6$–1.0, and $Bn = 0$–10. The specified ranges of these parameters were taken in accordance with the data on real drilling fluids and flow regimes in wells.

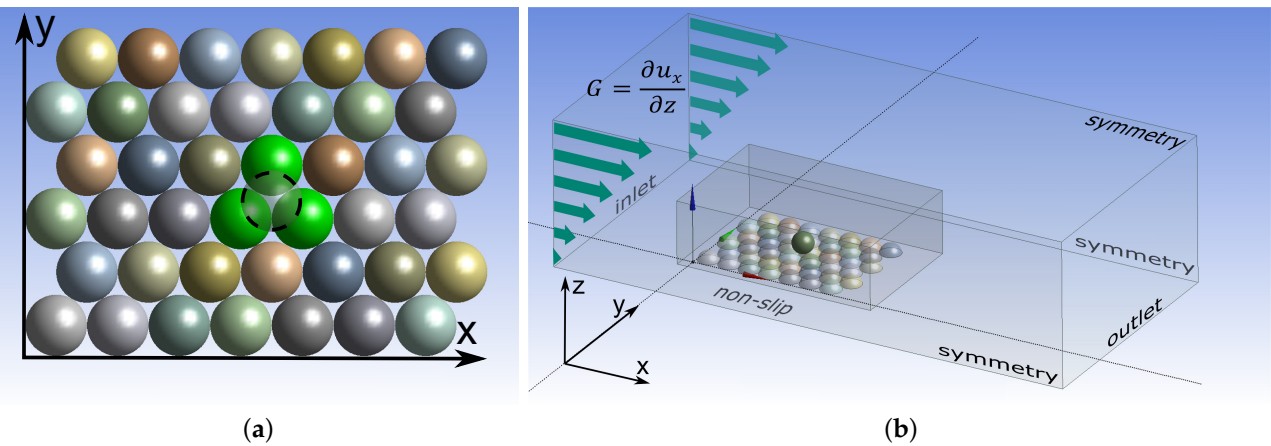

(**a**)                                                                                   (**b**)

**Figure 1.** (**a**) Hemisphere pattern on the substrate and the particle location above the lacuna between three hemispheres; (**b**) computational domain scheme.

The main integral force characteristics for a sphere located on a rough surface are the drag force $F_D = F_x$, the lift force $F_L = F_z$, and the torque $M = M_y$ relative to the axis passing through the center of the particle parallel to o–y. In engineering program complexes for cuttings transportation estimation [10], the dimensionless coefficients of the drag force $C_D$, the lift force $C_L$, and the torque $C_M$ determined with the following formulas are used:

$$C_D = F_D/F_S, C_L = F_L/F_S, C_M = 2M/F_Sd, \qquad (2)$$

where $F_S$ is the characteristic force acting on the diameter cross-section of the particle, and $F_S = 0.5u_p^2\rho\pi(d/2)^2$, where $u_p$ is the velocity in the flow at the center of the particle in the absence of the particle, which is the undisturbed velocity. The value of $u_p$ is actually determined via the local flow velocity near the rough wall, which even for a Newtonian fluid can differ significantly from the value $u_p^{smooth} = Gh_0$ (for $h_0$, see Section 3), as shown in [13]. Thus, the computation of the shear flow for the particle-free configuration was carried out in addition to each numerical computation of the particle on the rough surface.

## 3. Numerical Algorithm and Grid

The numerical algorithm is based on the finite volume method for an unstructured mesh. Laminar steady state fluid flow was simulated using the OpenFOAM CFD package (simpleFoam solver) [19]. The SIMPLE-C algorithm [20] was applied for the pressure correction procedure and grid placement with Rhie–Chow interpolation. The system of linear algebraic equations for the pressure correction equation was solved using an algebraic multigrid solver. A second-order linear upwind scheme was applied for the discretization of convective terms. The simulation was considered convergent when the residuals of velocity and pressure were less than $10^{-9}$ for Newtonian and power law fluids and less then $10^{-7}$ for Bingham and Herschel-Bulkley fluids. It is harder to achieve very low residuals for fluids with yield stress.

To correctly describe the viscosity behavior for small values of shear rate $\dot{\gamma}$, the regularization $\mu = \left[k\dot{\gamma}^n + \tau_y(1 - e^{-m\dot{\gamma}/G})\right]/\dot{\gamma}$ proposed in [21] by Papanastasiou was used, and the regularization parameter $m$ was taken as 1000.

The minimum distance from the center of the sphere to the plane on which the hemispheres are located is $h_0 = \sqrt{2/3}d \approx 0.816d$. Let us denote $\delta = h - h_0$, where $h$ is the distance from the center of the particle under study to the x–y plane. Thus, when $\delta = 0$, the particle touches the three hemispheres lying under it. The particle is located on the substrate after the fourth row of hemispheres along the x axis and after the third row along the y axis, as well as above the lacuna formed by the three hemispheres (Figure 1); the center of the sphere has the coordinates $(x_p, y_p, h)$. Assuming that the center of the first hemisphere is at the point $(0.5d, 0.5d, 0)$, we have $x_p = 4d$, $y_p = 0.5d(1 + 2\sqrt{3} + 1/\sqrt{3})$.

As shown in [13], in the chosen position, the structure of the laminar shear flow is fully formed over the rough surface. For the particle incipient motion from the substrate, the most interesting case is when the particle is located at the minimum distance $\delta = 0.005d$ from the rough surface, but for the purpose of modeling verification, other values of $\delta$ have also been considered.

The computational mesh was constructed in the ANSYS meshing package [22]. The computational domain was parallelepiped of dimensions $30d \times 15d \times 7d$ (Figure 1), in which a coarse mesh with a cell size of the order of $0.2d$ was constructed (Figure 2a). The bedload and the particle were surrounded by a small parallelepiped of $9d \times 6.8d \times 3d$ with a finer grid with a cell size of the order of $0.02d$ in the region itself and inflation layers near the particle surface (Figure 2b). The shapes of the elements were tetrahedrons and prisms. After the grid parameter adjustment, the surface size of the tetrahedra on a particle was chosen to be $0.005d$. The thickness of the particle's inflation layers was $0.005d$, the size of the tetrahedra in the small parallelepiped was $0.15d$, the size of the tetrahedra in the large parallelepiped was $0.4d$, and the total mesh size was 6 million cells. Computation on such a grid produced satisfactory results compared to the data of [13], but to increase the accuracy of simulation of the non-Newtonian fluid flow, the surface size of the elements on the particle was reduced to $0.003d$, and the thickness of the first cell in the inflation layers was reduced to $0.001d$ (Figure 2c), while the number of grid cells was increased to 19 million.

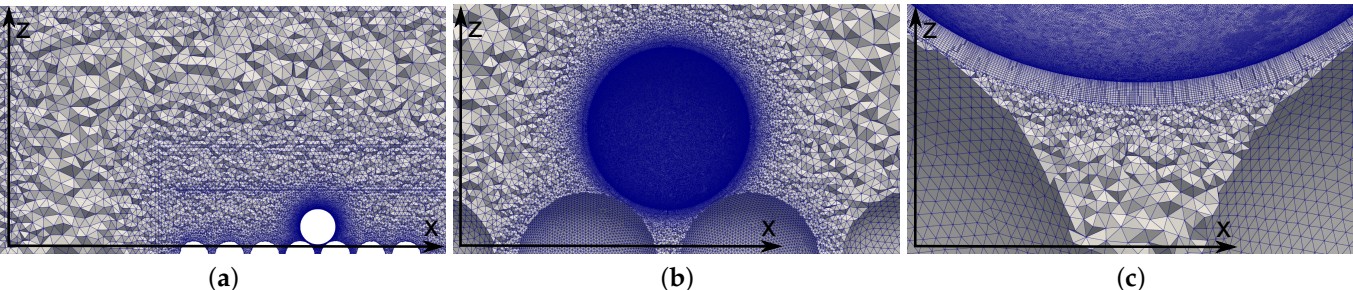

**(a)**          **(b)**          **(c)**

**Figure 2.** A grid in the x–z plane passing through the center of the sphere. (**a**) Inlet part of computational domain; (**b**) mesh around the sphere; (**c**) inflation layers around the sphere.

The incoming fluid flow was characterized by a constant shear $G$ and directed along the x axis; thus, the velocity at the inlet was $(u_x, u_y, u_z) = (G \cdot z, 0, 0)$. On the x–y plane and the hemispheres, a no-slip boundary condition was applied. On the side faces of the large parallelepiped, the symmetry boundary condition was used. A zero velocity gradient was set at the outlet. Figure 1 shows the geometry and boundaries.

The correctness of the computations was verified through the comparison of the Newtonian shear flow calculation results with the data presented in [13]. In [13], the authors validated the choice of the number of hemispheres for the formation of steady flow over a rough surface. The particle was at distances $\delta = 0.005, 0.1, 0.5, 1.0d$ from the rough surface; the Reynolds numbers were $Re_{SH} = 2, 10, 100$. Hydrodynamics simulations for the particle positions $\delta = 0.005, 0.1d$ did not reveal the lift force coefficient decrease as observed in [13] but demonstrated an increase in the $C_L$ by 16% compared with the results for the particle on the smooth wall [8]. For Reynolds numbers of 10 and 100, the difference between the $C_L$ coefficient with that calculated by Balachandar [13] did not exceed 5%. For the parameters $C_D$ and $u_p$, results consistent with the data of the article were obtained. The comparison is presented in Table 1.

A grid study was carried out. The regime with the highest demand on the grid flow of $Re_{SH} = 200$, $Bn = 10$, and $n = 0.6$ was chosen for testing. Control simulations were carried out on four grids: the basic grid and three finer grids. During grid construction, all linear values in the settings were changed by 1, 0.85, 0.7, and 0.5 times with respect to the basic grid. As can be seen from Table 2, the drag force and lift force behaved monotonically, while

the torque varied substantially. The deviations in the drag force, lifting force, and torque from the most detailed grid were −1.5%, −2.4%, and 8.5%, respectively.

**Table 1.** Comparison of simulation results for a particle on a rough surface in a shear flow of Newtonian fluid.

| $Re_{SH}$ | $\delta$ | Lee (2017) [13] | | Current Study | | Deviation | |
|---|---|---|---|---|---|---|---|
| | | $C_D$ | $u_p$ | $C_D$ | $u_p$ | $C_D, \%$ | $u_p, \%$ |
| 2 | 0.005 | 43.894 | 1.1 | 44.434 | 1.13 | 1.2 | 2.7 |
| | 0.1 | 33.157 | 1.3 | 34.451 | 1.305 | 3.9 | 0.4 |
| | 0.5 | 16.236 | 2.3 | 16.812 | 2.299 | 3.5 | 0.06 |
| | 1 | 10.34 | 3.5 | 10.264 | 3.496 | 0.7 | 0.1 |
| 10 | 0.005 | 11.018 | 5.6 | 11.258 | 5.678 | 2.2 | 1.4 |
| | 0.1 | 8.462 | 6.8 | 8.709 | 6.986 | 2.9 | 2.7 |
| | 0.5 | 4.53 | 11.9 | 4.628 | 12.105 | 2.2 | 1.7 |
| | 1 | 3.163 | 17.8 | 3.249 | 17.996 | 2.7 | 1.1 |
| 100 | 0.005 | 2.158 | 66.4 | 2.204 | 67.745 | 2.2 | 2 |
| | 0.1 | 1.763 | 80.3 | 1.825 | 81.215 | 3.5 | 1.1 |
| | 0.5 | 1.146 | 129 | 1.162 | 129.91 | 1.4 | 0.7 |
| | 1 | 0.918 | 181.3 | 0.933 | 181.965 | 1.6 | 0.4 |

**Table 2.** Grid study. $C_D$, $C_L$, and $C_M$ obtained using meshes with different refinement. Values in brackets are deviations in % from finest mesh.

| Settings Grid Refinement Ratio | Number of Cells $(10^6)$ | Effective Grid Refinement Ratio | $C_D$ | $C_L$ | $C_M \cdot (10^2)$ |
|---|---|---|---|---|---|
| 1 | 19.02 | 1 | 0.7170 (−1.5%) | 0.2752 (−2.4%) | −1.368 (8.5%) |
| 0.85 | 26.23 | 0.9 | 0.7205 (−1.0%) | 0.2777 (−1.5%) | −1.350 (7.0%) |
| 0.7 | 37.93 | 0.79 | 0.7238 (−0.6%) | 0.2802 (−0.7%) | −1.308 (3.7%) |
| 0.5 | 71.36 | 0.64 | 0.7279 (0.0%) | 0.2820 (0.0%) | −1.261 (0.0%) |

It should be noted that the chosen algorithm implies the construction of an unstructured mesh in a rather complex geometry. That is, the grids are no longer embedded. It is impossible to guarantee the same mesh quality even with a simultaneous proportional change in the linear parameters of the grid builder setting. Accordingly, it is also impossible to expect monotonic dependence of the solution when making the grid finer. We can only find the effective grid refinement ratio, which does not fully reflect the change in the average cell size from grid to grid. So, the application of Richardson extrapolation is quite difficult, and therefore, it is not possible to analyze the numerical error. Only qualitatively can we discuss the adequacy of the grid. Based on the deviations of the integral characteristics obtained on the basic grid compared to the most detailed grid, we consider the grid satisfactory.

On average, the simulation process took 4.5 days with eight cores parallelized using an Intel(R) Xeon(R) CPU E5-2658 v3 @ 2.20 GHz processor.

## 4. Results and Discussion

### 4.1. Force and Torque Coefficients

Let us consider the effect of rheology and the Reynolds number on the free-stream velocity on a particle center over a rough surface. In the case of a smooth wall, the velocity is given as follows: $u_p^{smooth} = Gh_0$. In the case of a rough wall, this value is distorted by the particle bed. The ratio of the free-stream velocity in front of the particle over a rough surface to the velocity over a smooth surface is denoted by

$$\theta(Re_{SH}, n, Bn) = u_p/u_p^{smooth}, \tag{3}$$

as shown in Figure 3. It is quite obvious that the particle bedding reduces the velocity of the free stream; thus, the value $\theta$ is always less than one. The $\theta$ value tends toward one with increases in the Reynolds number, Bingham number, and decreases in the power law index. The decrease in the power law index and increase in the Reynolds number decrease the apparent viscosity; hence, this results in the decreasing influence of the particle bedding. It may seem that increasing the Bingham number *Bn* should lead to an increase in the yield stress $\tau_y$ and apparent viscosity. However, since the Bingham number *Bn* is included in the denominator of the shear Reynolds number $Re_{SH}$, the opposite effect is observed. An increase in the Bingham number *Bn* leads not only to a nominal increase in the yield stress $\tau_y$ but also to an increase in the shear rate *G* of the free stream. Of course, increasing the yield stress while fixing other parameters would lead to an increase in the apparent viscosity, but the introduced dimensionless parameters make the analysis slightly more complex. Thus, the effect of the Bingham number is inverted by the introduced dimensionless parameters.

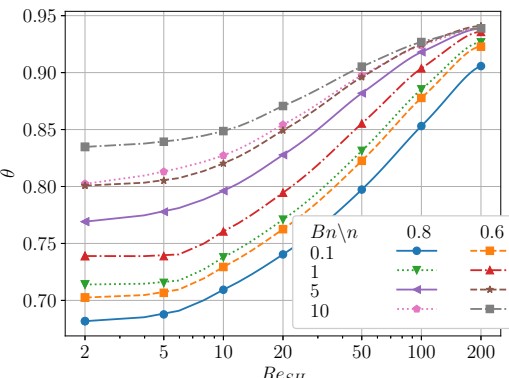

**Figure 3.** Dependence of the free-stream velocity coefficient $\theta$ on the Reynolds number $Re_{SH}$, Bingham number *Bn*, and power law index *n*.

It was shown in [9] that as the Bingham number *Bn* increases and the power law index *n* decreases, the drag coefficient $C_D$ decreases for a smooth wall. The same behavior is true for a rough wall (Figure 4a). A decrease in the power law index *n* decreases the fluid viscosity; thus, the drag force $C_D$ decreases. As noted above, the effect of the Bingham *Bn* number's influence is inverted, so the drag force $C_D$ decreases as the Bingham number increases. Nevertheless, taking into account the decreasing $C_D$ coefficient, we can preliminarily conclude that roughness for pseudoplastic fluids with significant yield stress prevents particle motion at the surface.

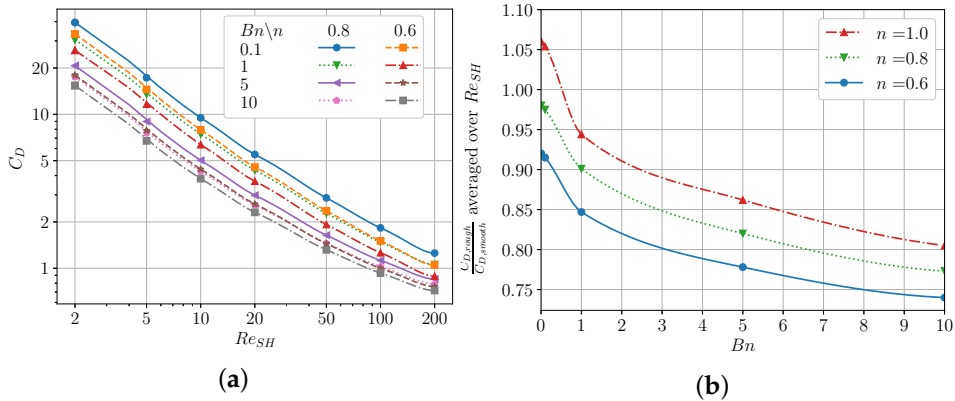

**(a)**　　　　　　　　　　　　　　　**(b)**

**Figure 4.** (**a**) Dependence of the drag force coefficient $C_D$ on the Reynolds number $Re_{SH}$, Bingham number *Bn*, and power law index *n*. (**b**) Influence of the rough wall on the drag force depending on rheological parameters *n* and *Bn*. Ratio $\frac{C_{D,rough}}{C_{D,smooth}}$ averaged over $Re_{SH}$.

Let us consider the effect of roughness depending on the rheological properties of the fluid on the drag force. Since there are three variables in the problem, for simplicity of presentation we averaged the ratio of the drag force on a rough wall to the force on a smooth wall over the Reynolds number, thereby leaving the influence of the rheological properties only. The same was performed for the lifting force and the torque acting on the particle. As shown earlier in [13] for a Newtonian fluid, a rough surface increases the resistance in comparison with a smooth wall by up to 10%. This conclusion was confirmed by our simulations; we found that, on average, the drag force was 6% higher over a rough wall (Figure 4b). As the non-Newtonian properties of the fluid increase, the drag force coefficient decreases over the rough surface compared to the smooth wall. Thus, the effect of the rough wall on the drag force is opposite to that of the non-Newtonian fluid rheology. Apparently, this is due to the large velocity gradients caused by the particle bed, which reduce viscosity and drag. It should be noted that applying correlations for smooth walls will overestimate the drag force.

The deviation analysis for determining the drag force over a rough surface using the simulations for smooth walls is given below. The correlation for a spherical particle at a distance from the wall as a function of the Reynolds number and distance to the wall was developed by Zeng et al. [8]. The reduced expression when the particle touches the wall is shown below:

$$C_D = \frac{24 \cdot 1.3255}{Re}\left(1 + 0.104 Re^{0.753}\right),\tag{4}$$

where the particle Reynolds number is $Re = \rho G d^2/2\mu$. Using Formula (4) gives the average absolute error $25 \pm 12\%$, and the total error range is from $-53\%$ to $48\%$. In [9], data on the drag force for the case of the Herschel–Bulkley fluid and smooth wall are presented. The average absolute error when using this result is $19 \pm 7\%$, while the relative error varies from $-3\%$ to $62\%$. As an example, Table 3 compares the drag coefficient obtained in this work for a rough wall to that in [9] and from Zeng's correlation for the case of $n = 0.6$, with $Bn = 10$ as the case with the highest expression. As expected, both correlations noticeably overestimate the drag force.

**Table 3.** Drag force, lift force, and torque coefficient for $Bn = 10$ and $n = 0.6$ simulated for rough and smooth walls, which were calculated from correlations. Values in brackets are deviations in % from rough wall.

| $Re_{SH}$ | $C_D$ | | | $C_L$ | | | $C_M$ | |
|---|---|---|---|---|---|---|---|---|
| | **Rough** | **Smooth** | **Zeng** [1] | **Rough** | **Smooth** | **Zeng** [2] | **Rough** | **Smooth** |
| 2 | 15.39 | 24.95 (+62%) | 18.69 (+21%) | 1.712 | 1.906 (+11%) | 2.683 (+57%) | 2.306 | 3.566 (+55%) |
| 5 | 6.725 | 10.28 (+52%) | 8.585 (+28%) | 1.306 | 1.301 (−0.3%) | 1.802 (+38%) | 0.856 | 1.245 (+46%) |
| 10 | 3.821 | 5.673 (+49%) | 5.055 (+32%) | 1.09 | 1.114 (+2.2%) | 1.330 (+22%) | 0.365 | 0.579 (+59%) |
| 20 | 2.309 | 3.221 (+40%) | 3.169 (+37%) | 0.936 | 0.895 (−4.3%) | 0.980 (+5%) | 0.124 | 0.215 (+74%) |
| 50 | 1.318 | 1.658 (+26%) | 1.895 (+44%) | 0.851 | 0.689 (−20%) | 0.655 (−23%) | −0.011 | 0.009 (+177%) |
| 100 | 0.931 | 1.119 (+20%) | 1.379 (+48%) | 0.63 | 0.644 (+2.2%) | 0.483 (−23%) | −0.026 | −0.039 (+50%) |
| 200 | 0.717 | 0.805 (+12%) | 1.053 (+47%) | 0.275 | 0.45 (+63%) | 0.356 (+29%) | −0.014 | −0.039 (+184%) |

[1] using Formula (4) from [8]. [2] using Formula (5) from [8].

The characteristic of the rheology dependence of the lifting force coefficient $C_L$ is the same as that of the drag force $C_D$, i.e., it decreases with the increase in the Bingham number $Bn$ and decrease in the power law index $n$ (Figure 5a). The lifting force coefficient $C_L$ also decreases with the increase in the Reynolds number $Re_{SH}$ but not as fast as the drag coefficient. It is interesting that at Reynolds numbers of $Re_{SH} = 50$–200, the dependence on rheology becomes noticeably smaller, and the curves become close to each other.

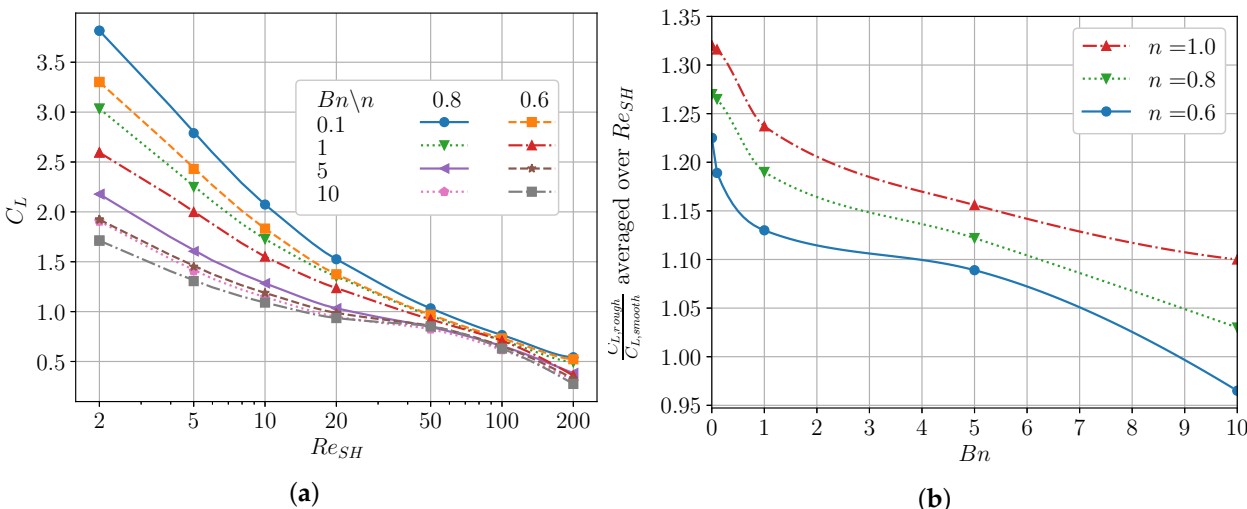

**Figure 5.** (**a**) Dependence of the lift force coefficient $C_L$ on the Reynolds number $Re_{SH}$, Bingham number $Bn$, and power law index $n$. (**b**) Influence of the rough wall on the lift force depending on rheological parameters $n$ and $Bn$. Ratio $\frac{C_{L,rough}}{C_{L,smooth}}$ averaged over $Re_{SH}$.

As well as the drag force, we analyzed the effect of roughness on the lift force. As can be seen, the lifting force for the Newtonian fluid is on average up to 30% higher in the case of the rough wall (Figure 5b). For the non-Newtonian fluid, as well as in the case of drag force, decreasing the power law index $n$ and increasing the Bingham number $Bn$ leads to a decrease in the effect of the rough wall. The lift force coefficient $C_L$ for almost all of the considered cases is greater than the $C_L$ obtained for the particle on a smooth wall. This effect is connected, in particular, to the increase in hydrodynamic pressure value $p$ and its redistribution. The visualization in Figure 6 is made in such a way that the bottom point of the sphere can be taken as zero pressure. In both cases, the negative pressure region occupies a slightly larger area; thus, there is a deflection region above the upper point of the particle. In the case of the rough wall, the negative pressure region increases, thus spreading slightly upstream on the sphere. As a result, the lift force acting on the sphere over the sedimentation layer increases compared to the particle on the smooth wall. To visually estimate the change in pressure distribution during the transition to a rough surface, Figure 6 shows the isolines corresponding to $p = 30$ in the frontal zone of the particle and $p = -40$ in the rarefaction region behind the sphere.

In a study by Zeng et al. [8], the formula for describing the lift force coefficient $C_L$ for a particle over a smooth surface is given. In the case of a particle touching the surface, it can be written as follows:

$$C_L = \frac{3.663}{(Re^2 + 0.1173)^{0.22}}, \tag{5}$$

where the Reynolds number $Re$ is determined from the flow velocity at the center of the particle. Let us apply it for the case of the rough wall and assess the difference. Using Formula (5) gives an average difference of $22 \pm 9\%$, with the whole difference ranging from $-36\%$ to $56\%$. An estimation of the lifting force $C_L$ for a smooth wall and a Herschel–Bulkley fluid from [9] gives an average difference of $14 \pm 12\%$ and a full difference range from $-24\%$ to $63\%$. As an example of the largest deviation of the estimates of $C_L$ for a smooth wall from a rough wall, we give this example for $n = 0.6$ and $Bn = 10$ (Table 3). If for most regimes the estimates for a smooth wall give an underestimated value of $C_L$, i.e., the deviation is negative; then here, both Zeng's Formula (5) and the simulation in [9] can give errors of different signs.

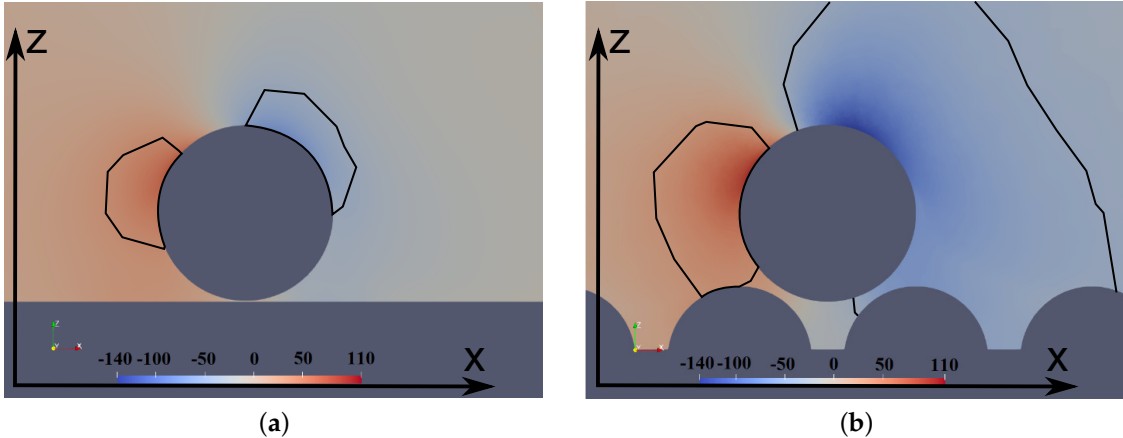

(a)                                      (b)

**Figure 6.** Distribution of dimensionless pressure in the case of Herschel–Bulkley fluid flow with parameters $n = 0.8$ and $Bn = 1$ around a sphere on (**a**) a smooth and (**b**) a rough surface. Isolines (black solid curves) show $p = 30$ in the frontal zone of the particle and $p = -40$ in the rarefaction region behind the sphere.

As for the drag and lift coefficients, $C_M$ decreases with the decrease in the power law index $n$, thereby increasing the Bingham $Bn$ and Reynolds numbers $Re_{SH}$. It should be noted that $C_M$ decreases very rapidly with the increase in the Reynolds number, so that at $Re_{SH} > 50$, the value of $C_M$ becomes close to zero and even negative (Figure 7a). This can be explained by the observation that with the increase in yield stress, the torque on the particle decreases due to the velocity magnitude growth at the front of the sphere, with the velocity vector directed downward to the lacuna between the three hemispheres. As a result, a torque opposite to the clockwise rotation of the particle is created. When the Bingham number reaches 5–10 and $n$ is decreased to 0.8–0.6, the contribution of this torque becomes so significant that negative values of the integral torque $M$ acting on the sphere are recorded.

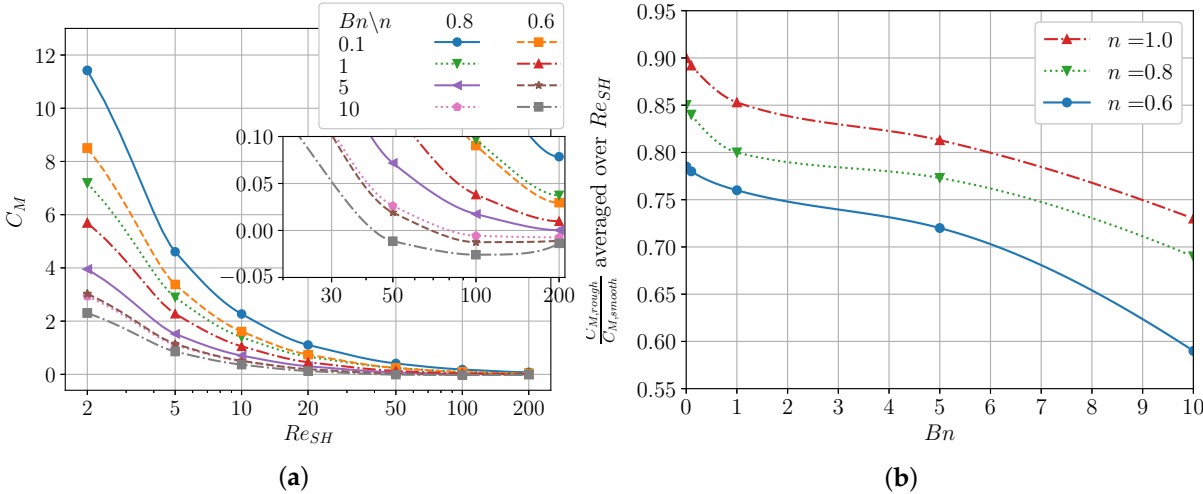

(a)                                      (b)

**Figure 7.** (**a**) Dependence of the torque coefficient $C_M$ on the Reynolds number $Re_{SH}$, Bingham number $Bn$, and power law index $n$. (**b**) Influence of the rough wall on the torque depending on rheological parameters $n$ and $Bn$. Ratio $\frac{C_{M,rough}}{C_{M,smooth}}$ averaged over $Re_{SH}$.

Comparing the Bingham fluid flow visualization around the particle on the rough surface at $Re_{SH} = 200$ and different $Bn$ values, it was observed that at a low value of $\tau_y$ ($Bn = 0.1$; Figure 8a), there were streamlines behind the particle passing from the front through the lacuna between the three hemispheres. As a result, on the lower back side of the particle for $Bn = 0.1$, we have an upward velocity, thus creating a reverse torque $\widetilde{M}$. When

the particle was located in the shear flow with $Bn = 10$ (Figure 8b), a stagnation zone was formed in the lacuna, through which a small volume of liquid passes; as a result, the velocity on the back side of the particle was entirely directed toward the surface. It might seem that the torque $\widetilde{M}$ should decrease the total parameter $M$ for $Bn = 0.1$ compared to $Bn = 10$, but this does not happen due to the fact that the magnitude of the velocity behind the particle near the surface is two orders of magnitude lower than the velocity induced at the sphere front.

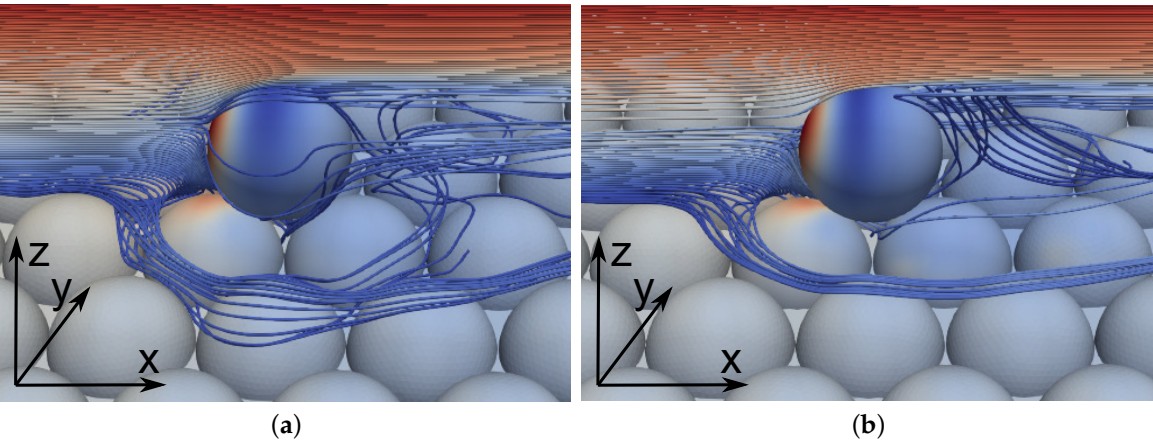

**(a)**  **(b)**

**Figure 8.** Streamlines for Bingham fluid with (a) $Bn = 0.1$ and (b) $Bn = 10$.

The torque coefficient $C_M$ of a particle on a rough surface in all considered simulation cases was less than that for a particle on a smooth wall. The fluid that flowed over the particle created a positive torque acting on the particle. Correspondingly, the fluid flowing under the particle created a negative torque acting on the particle. In the case of a rough surface, the lacuna under the particle reduced the total torque compared to a smooth surface, because more fluid is able to flow under the particle. The smooth wall torque was 10% greater than the torque on the rough wall in the case of the Newtonian fluid, and this difference increased with the increase in the Bingham number $Bn$ and decrease in the power law index $n$ (Figure 7b). The average difference between the torques on the rough wall and smooth wall was around 51%.

### 4.2. Approximation of Forces and Torque

#### 4.2.1. The Drag Force Coefficient $C_D$

The dependence of $C_D$ on $Re_{SH}$ in the bilogarithmic scale is almost linear, so $C_D(Re)$ should be well described by a power function. An approximation in the form of Formula (6) is sought, which is similar to the known Schiller–Naumann correlation [23] for an unbounded medium and the Zeng et al. correlation (Formula (4)). Coefficients $A$, $B$, and $C$ of Formula (6) are assumed to depend on the power law index $n$ and Bingham number $Bn$ and are approximated in the form of Functions (7), (8), and (9), respectively, with appropriate coefficients in Table 4.

$$C_D = A\left(1 + B \cdot Re_{SH}^C\right) \tag{6}$$

$$A = exp\left(a_0 + a_1 Bn + a_2 Bn^{3/2} + a_3 e^{Bn/a_4} + a_5 e^{-Bn} + a_6/\sqrt{n} + a_7 \ln n/n + a_8/n\right) \tag{7}$$

$$B = exp\left(b_0 + b_1 n + (b_2 n^2 + b_3 n + b_4)Bn + (b_5 n^2 + b_6 n + b_7)Bn^{3/2} + (b_8 n + b_9)\sqrt{Bn}\right) \tag{8}$$

$$C = c_0 + c_1 Bn + c_2 n + c_3 Bn^2 + c_4 n^2 + c_5 Bn \cdot n + c_6 Bn^3 + c_7 n^3 + c_8 Bn \cdot n^2 + c_9 Bn^2 n \tag{9}$$

The approximation (6) of $C_D$ gives an average error of $2.5 \pm 2\%$, with the maximum not exceeding 9%. The distribution of the error is shown in Figure 9a.

**Table 4.** Coefficients of approximation formula for the drag force coefficient $C_D$ (Formulas (6)–(9)).

| $i$ | $a_i$ | $b_i$ | $c_i$ |
|---|---|---|---|
| 0 | −112.56593 | −4.2837156 | 0.6749968 |
| 1 | 15.401215 | 2.2809931 | 0.023630196 |
| 2 | −2.0707175 | −0.024535228 | 0.33005892 |
| 3 | 156.19872 | 0.92007397 | −0.014190415 |
| 4 | −12.22632973 | −0.54858251 | −0.5982012 |
| 5 | 2.0488369 | 0.14570464 | 0.11350501 |
| 6 | −147.1652 | −0.37801647 | $9.341168 \cdot 10^{-4}$ |
| 7 | 32.832341 | 0.18962019 | 0.19515689 |
| 8 | 105.83532 | −2.1212564 | −0.035606101 |
| 9 | – | 1.1468118 | $-3.177588 \cdot 10^{-3}$ |

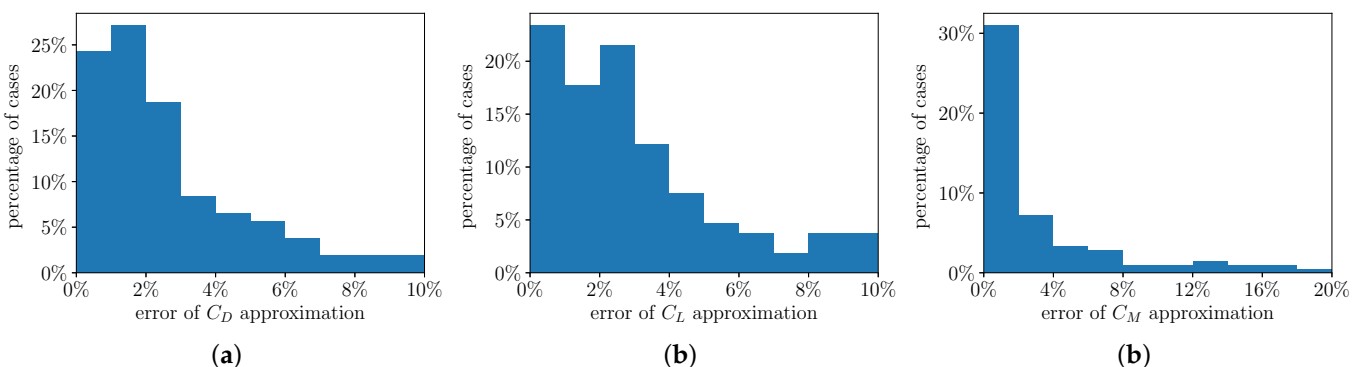

(**a**)      (**b**)      (**b**)

**Figure 9.** Histogram of approximation error of (**a**) $C_D$ for Formula (6), (**b**) $C_L$ for Formula (10), and (**c**) $C_M$ for Formula (11).

### 4.2.2. The Lift Force Coefficient $C_L$

The approximation formula for the lift force coefficient $C_L$ is found in the same way as for the drag coefficient $C_D$, i.e., we first find the universal dependence on the Reynolds number $Re_{SH}$ and then approximate the coefficients. The dependence of the lift force coefficient $C_L$ on the Reynolds number $Re_{SH}$ is more complex compared to the drag force coefficient $C_D$. Using an approximation formula for $C_L$ in a Newtonian fluid, similar to Formula (5) and with arbitrary coefficients gives a large error, and the dependence of the free coefficients on $n$ and $Bn$ is highly nonlinear. To approximate the lift force coefficient $C_L$, expression (10) with coefficients $A$, $B$, $C$, and $D$, was chosen. These coefficients are described in the same way as in expression (9). The coefficients for them are given in Table 5.

$$C_L = A + B \cdot Re_{SH} + \frac{C}{Re_{SH}} + \frac{D}{Re_{SH}^2} \tag{10}$$

**Table 5.** Coefficients of approximation formula for the lift force coefficient $C_L$ (Formula (10)).

| $i$ | $a_i$ | $b_i$ | $c_i$ | $d_i$ |
|---|---|---|---|---|
| 0 | 0.91355258 | $-2.627284 \cdot 10^{-3}$ | −0.73318047 | 33.637812 |
| 1 | 0.073666872 | $-7.489556 \cdot 10^{-4}$ | −3.4818427 | 3.3643489 |
| 2 | −0.064727027 | $1.449468 \cdot 10^{-3}$ | 25.874692 | −154.9099 |
| 3 | $-7.389389 \cdot 10^{-3}$ | $8.302351 \cdot 10^{-5}$ | 0.8370182 | −0.93646073 |
| 4 | 0.047693492 | $-1.604971 \cdot 10^{-3}$ | −14.962358 | 179.25061 |
| 5 | −0.094994519 | $6.396882 \cdot 10^{-4}$ | −2.1984015 | 4.0141986 |
| 6 | $2.533976 \cdot 10^{4}$ | $-3.407309 \cdot 10^{-6}$ | −0.05002062 | 0.059314397 |
| 7 | 0.15321461 | $-1.169344 \cdot 10^{-4}$ | 3.6677747 | −73.136321 |
| 8 | −0.016809973 | $8.652267 \cdot 10^{-5}$ | 0.84560024 | −1.4517147 |
| 9 | $6.579130 \cdot 10^{-3}$ | $-4.242128 \cdot 10^{-5}$ | 0.051772368 | −0.141001253 |

The approximation (10) of $C_L$ gives an average error of $3 \pm 2.5\%$, with the maximum not exceeding 10%. The distribution of the error is shown in Figure 9b.

### 4.2.3. The Torque Coefficient $C_M$

In contrast to the coefficients $C_D$ and $C_L$, no example of an approximation formula for the torque coefficient $C_M$ was found. Since the torque coefficient $C_M$ decreases quickly with the increase in the Reynolds number $Re_{SH}$, the proposed expression consists of a constant and the sum of slowly increasing and rapidly decreasing functions:

$$C_M = A + B\sqrt{Re_{SH}} + \frac{C}{Re_{SH}} + D \cdot exp(-Re_{SH}). \tag{11}$$

The approximation of the coefficients A, C, and D are given in Formulas (12)–(14). The function $B$ for the torque is described in the same way as in expression (9). The coefficients for them are given in Table 6.

$$A = \frac{a_0 + a_1 Bn + a_2 Bn^2 + a_3 \ln n}{1 + a_4 Bn + a_5 Bn^2 + a_6 \ln n + a_7 \ln^2 n} \tag{12}$$

$$C = \frac{c_0 + c_1 Bn + c_2 Bn^2 + c_3 ln(n) + c_4 \ln^2 n}{1 + c_5 Bn + c_6 Bn^2 + c_7 Bn^3 + c_8 \ln n} \tag{13}$$

$$D = d_0 + d_1 \ln Bn + d_2 n + d_3 \ln^2 Bn + d_4 n^2 + d_5 n \ln Bn + d_6 \ln^3 Bn + d_7 n^3 + d_8 n^2 \ln Bn + d_9 n \ln^2 Bn \tag{14}$$

The approximation (11) of $C_M$ gives an average error of $3.2 \pm 3\%$, with the maximum not exceeding 19%. The distribution of the error is shown in Figure 9c.

**Table 6.** Coefficients of approximation formula for the torque coefficient $C_M$.

| $i$ | $a_i$ | $b_i$ | $c_i$ | $d_i$ |
|---|---|---|---|---|
| 0 | $2.864166 \cdot 10^{-3}$ | 0.013313068 | 32.20391 | 178.72147 |
| 1 | $-1.247596 \cdot 10^{-2}$ | $-4.563881 \cdot 10^{-3}$ | 1.315694 | 0.46516483 |
| 2 | $-6.825300 \cdot 10^{-4}$ | 0.061773494 | $-0.03373$ | $-710.19627$ |
| 3 | $7.696991 \cdot 10^{-2}$ | $-1.202564 \cdot 10^{-4}$ | 41.63685 | $-0.014039359$ |
| 4 | $8.356225 \cdot 10^{-2}$ | $-0.17901219$ | 27.47855 | 916.98635 |
| 5 | $7.686759 \cdot 10^{-3}$ | 0.013726357 | 0.738348 | $-0.39068685$ |
| 6 | 3.0697927 | $-7.9884 \cdot 10^{-7}$ | $-0.05171$ | $-2.11435 \cdot 10^{-5}$ |
| 7 | 3.2006677 | 0.10828134 | 0.002346 | $-387.41501$ |
| 8 | – | $-9.178116 \cdot 10^{-3}$ | 0.101589 | 0.15712562 |
| 9 | – | $1.505406 \cdot 10^{-4}$ | – | $-2.513205 \cdot 10^{-43}$ |

### 4.3. Particle Incipient Motion Conditions

Let us first consider the incipient motion condition of the spherical particle. In the three-dimensional case, when the particle is located above the lacuna at the minimum distance, it touches the three lower hemispheres at points *A*, *B*, and *C* (Figure 10a). We place the center of the coordinates at the point where the hemispheres touch, as shown in Figure 10a. Since the flow is directed along the x axis, it makes sense to consider only the rolling condition at points *A* and *B*. For this, we need to write down the equation for the torque with respect to the contact points. Let us denote *D* as the center of the particle and *E* as the center of the circle of radius *R* passing through the centers of the hemispheres (Figure 10b). Consider that the integral forces $F_D$ and $F_L$ are applied at the point *D*, and the torque $\vec{M}$ has the components $(M_x, M_y, M_z)$, of which only $M_y \equiv M$ is relevant. In addition, there is a gravity force $\vec{F}_g = (0, 0, -\rho_p g V)$ applied at point *D*, where $V = (4/3)\pi(d/2)^3$ is the particle volume, $\rho_p$ is the particle material density, and the buoyancy force is $\vec{F}_A = (0, 0, \rho g V)$. From geometrical considerations, $R = d/\sqrt{3}$, $ED = H = \sqrt{2/3}d$, and $OE = d/2\sqrt{3}$. Point *A* has coordinates $(\delta_x, \delta_y, \delta_z)$, where $\delta_x = d/4$,

$\delta_y = d/4\sqrt{3}$, and $\delta_z = d/\sqrt{6}$. Point $B = (0, d/\sqrt{3}, d/\sqrt{6})$. Let us denote the torque of forces $\overrightarrow{M}_A$ of the particle with respect to point $A$:

$$\overrightarrow{M}_A = \overrightarrow{M} + \overrightarrow{a} \times \overrightarrow{F} + \overrightarrow{a} \times \left( \overrightarrow{F}_g + \overrightarrow{F}_A \right), \tag{15}$$

where $\overrightarrow{F} = (F_D, F_y, F_L)$, $F_y$ is the force acting on the particle perpendicular to the fluid flow, and $\overrightarrow{a} \equiv \overrightarrow{AD} = (-d/4, d/4\sqrt{3}, d/\sqrt{6})$. Since we are interested in a particle rolling out of the lacuna in the flow direction, we consider the condition for the y component of the torque $\overrightarrow{M}_A$, which, after the appropriate calculations, has the form

$$M + \frac{d}{4}F_L + \frac{d}{\sqrt{6}}F_D + \frac{d}{4}(\rho - \rho_p)Vg \geq 0. \tag{16}$$

Then, using representation (3), Formula (2), and considering $F_S = 0.5u_p^2\rho\pi(d/2)^2 = (1/3)\theta^2 G^2 d^2 \rho\pi(d/2)^2$, we obtain

$$G^2 d^2 \theta^2 \rho \left( C_M + \frac{1}{2}C_L + \sqrt{\frac{2}{3}}C_D \right) \geq (\rho_p - \rho)dg. \tag{17}$$

The dimension of the right- and left-hand sides of the inequality is stress. The characteristic local shear stress on the wall in the vicinity of the particle can be estimated as follows:

$$\tau_w = \mu \dot{\gamma}_{local}^n + \tau_y = k \left( \frac{u_p}{h_0} \right)^n + \tau_y = kG^n \left( \theta^n + Bn \right). \tag{18}$$

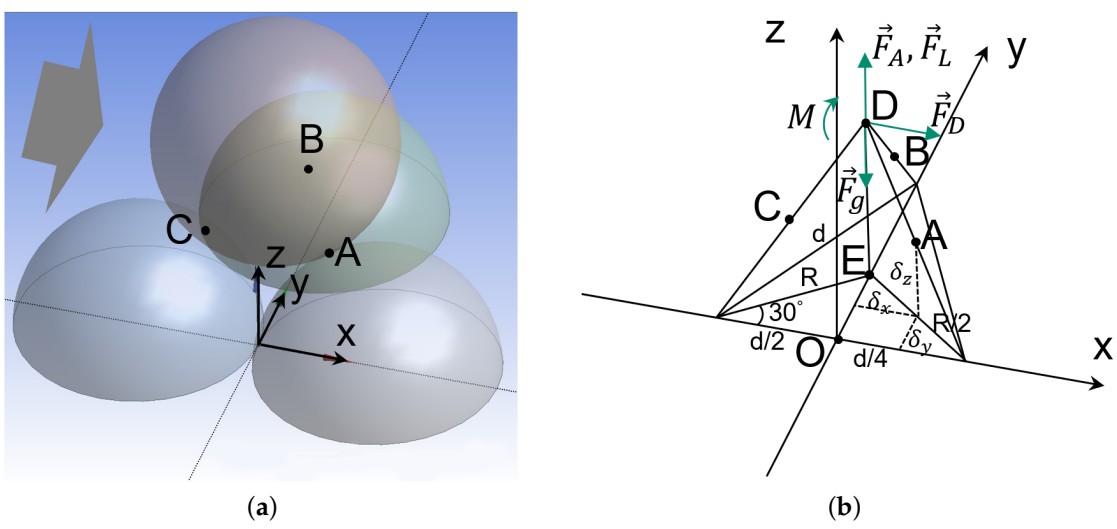

<div align="center">(<b>a</b>)                    (<b>b</b>)</div>

**Figure 10.** (**a**) Points of contact of a particle with hemispheres and (**b**) the scheme of action of the force and torque vectors.

Inequality (17) can be transformed to a dimensionless form by dividing each part of it by $\tau_w$:

$$\frac{\tau_w}{(\rho_p - \rho)dg} \geq \frac{1}{\text{Re}_u \left( \sqrt{\frac{3}{2}}C_M + \frac{1}{2}\sqrt{\frac{3}{2}}C_L + C_D \right)}, \tag{19}$$

Here, we denote by $\text{Re}_u$ the Reynolds number, which is constructed based on the local undisturbed flow velocity $u_p$:

$$\text{Re}_u = \sqrt{\frac{2}{3}} \frac{\rho G^{2-n}d^2\theta^2}{k(\theta^n + Bn)} = \sqrt{\frac{2}{3}} \frac{(1 + Bn)\theta^2}{(\theta^n + Bn)} \text{Re}_{SH}. \tag{20}$$

Note that in inequality (19), the left-hand side is the Shields number for the particle, and $\tau_B = \tau_w/(\rho_p - \rho)dg$. The critical value of $\tau_B$, at which the particle begins to roll out of the lacuna, is reached when the inequality turns into the equality

$$\tau_B = \tau_R \equiv \left\{ \mathrm{Re}_u \left( \sqrt{\frac{3}{2}} C_M + \frac{1}{2}\sqrt{\frac{3}{2}} C_L + C_D \right) \right\}^{-1}, \tag{21}$$

where we denote by $\tau_R$ the critical rolling Shields number.

Let us consider next the torque $\overrightarrow{M}_B$ in relation to the point $B$:

$$\overrightarrow{M}_B = \overrightarrow{M} + \overrightarrow{b} \times \overrightarrow{F} + \overrightarrow{b} \times \left( \overrightarrow{F}_g + \overrightarrow{F}_A \right), \tag{22}$$

where $\overrightarrow{b} \equiv \overrightarrow{BD} = (0, -d/2\sqrt{3}, d/\sqrt{6})$. The condition for the particle's rotation around the axis passing through $B$ parallel to the y axis is

$$M + \frac{d}{2}\sqrt{\frac{2}{3}} F_D \geq 0, \tag{23}$$

and is always satisfied in the range of parameters under study. Thus, for the beginning of the particle rolling out of the lacuna, it is only necessary to fulfill the condition (19).

The condition of complete detachment of the particle from the substrate and entrainment into the flow has a similar form for points $A$ and $B$ in terms of forces:

$$F_L + \left| \overrightarrow{F}_A \right| - \left| \overrightarrow{F}_g \right| \geq 0. \tag{24}$$

Using transformations similar to those performed with torque, we obtain

$$\tau_B \geq \frac{1}{\frac{1}{2}\sqrt{\frac{3}{2}} C_L \cdot \mathrm{Re}_u} \equiv \tau_L. \tag{25}$$

Thus, particle detachment is achieved when the Shields number reaches the critical value $\tau_L$.

It is undoubtedly important to separately determine the drag force, lift force, and torque coefficients, but in an actual simulation, similar to the mechanistic approach, these quantities are included together in the final expressions of particle motion. The critical shear stresses $\tau_R$ and $\tau_L$ have the same behavior as in the case of the smooth wall. The critical stresses decrease with the Reynolds number $Re_{SH}$ and increase with the increase in non-Newtonian properties. Furthermore, it should be noted that the critical stresses strongly depend on the rheological parameters $n$ and $Bn$ with small Reynolds numbers $Re_{SH}$, but with an increase in $Re_{SH}$, the dependence on rheology significantly decreases or becomes negligible (Figure 11). The more non-Newtonian properties a fluid has, the greater the stress that must be applied to initiate movement or lift a particle off from other settled particles.

The increasing level of non-Newtonian properties leads to decreases in the drag force coefficient $C_D$, lift force $C_L$, and torque $C_M$, which in turn increase the critical stresses for particle motion initiation and detachment from the sediment bed. Thus, in a non-Newtonian fluid, a higher local Reynolds number and shear rate are required to initiate particle motion and detachment in horizontal flow. Let us discuss the difference in critical shear stresses between smooth and rough walls. The average difference (underestimation) in the critical wall stress $\tau_R$ for the incipient motion between smooth and rough walls was $10 \pm 8\%$, while the maximum difference reached 35%. To detach a particle from a rough wall, an average of $12 \pm 9\%$ more stress $\tau_L$ is required but not more than 47%. The difference can be either positive or negative. The differences in $\tau_R$ and $\tau_L$ between the smooth and rough wall were noticeably smaller than the differences in the coefficients $C_D$, $C_L$, and $C_M$, since the latter were included in the expressions as a combination of each other and of

the $Re_u$ together. Zeng et al.'s [8] correlations (Equations (4) and (5)) can also be applied to calculate the critical stresses $\tau_R$ (Equation (21)) and $\tau_L$ (Equation (25)) for the incipient motion and detachment. Unfortunately, the deviation in this case was significantly higher compared to the data [9] for a smooth wall with a Herschel–Bulkley fluid. For motion initiation, the average deviation was $43 \pm 30\%$, and for detachment, it was $30 \pm 17\%$. Table 7 summarizes the values of the $\tau_R$ and $\tau_L$ coefficients for the smooth and rough surfaces for $n = 0.6, Bn = 10$ as one of the regions of highest deviation.

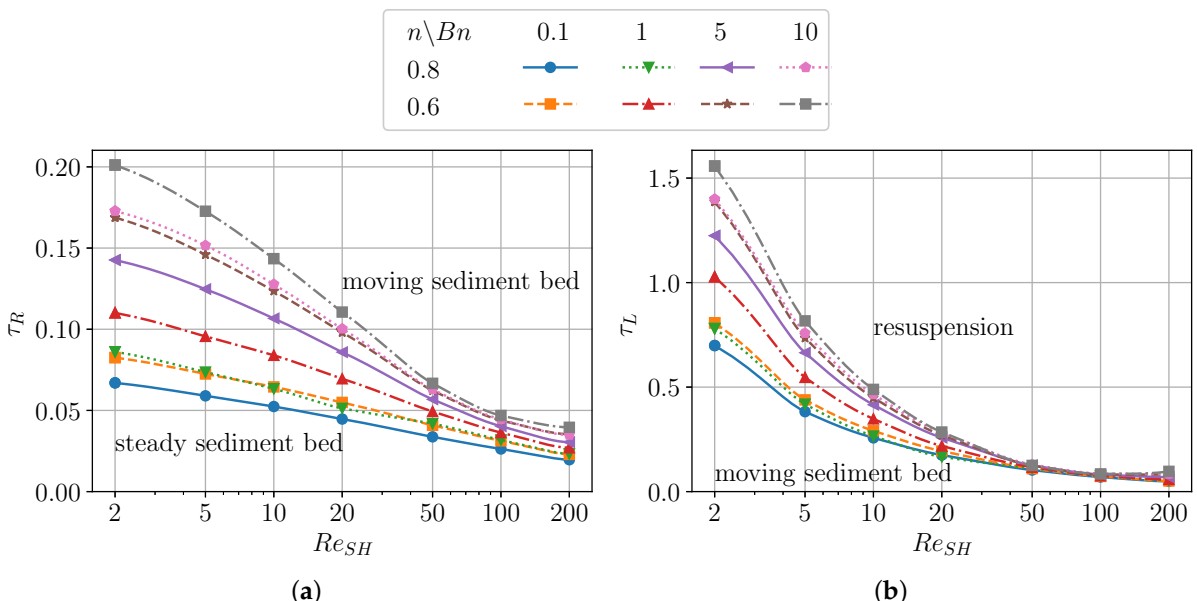

(a)  (b)

**Figure 11.** Dependence of critical Shields numbers for (**a**) rolling $\tau_R$ and (**b**) detachment from a rough surface $\tau_L$ on the Reynolds number $Re_{SH}$, Bingham number $Bn$, and power law index $n$.

**Table 7.** Comparison of simulation results for a particle on a rough surface in a shear flow of a Newtonian fluid.

| $Re_{SH}$ | $\tau_R$ | | | $\tau_L$ | | |
|---|---|---|---|---|---|---|
| | **Rough** | **Zeng** [1] | **Smooth** | **Rough** | **Zeng** [2] | **Smooth** |
| 2 | 0.2011 | 0.1979 (−2%) | 0.1305 (−35%) | 1.5576 | 0.9939 (−36%) | 1.3994 (−10%) |
| 5 | 0.1727 | 0.1578 (−9%) | 0.1220 (−28%) | 0.8169 | 0.5918 (−28%) | 0.8196 (0.3%) |
| 10 | 0.1435 | 0.1256 (−12%) | 0.1054 (−26%) | 0.4891 | 0.4011 (−18%) | 0.4787 (−2%) |
| 20 | 0.1106 | 0.0949 (−14%) | 0.0888 (−20%) | 0.2850 | 0.2720 (−5%) | 0.2978 (5%) |
| 50 | 0.0667 | 0.0600 (−9%) | 0.0644 (−4%) | 0.1253 | 0.1628 (30%) | 0.1548 (24%) |
| 100 | 0.0469 | 0.0417 (−11%) | 0.0428 (−9%) | 0.0847 | 0.1105 (30%) | 0.0829 (−2%) |
| 200 | 0.0396 | 0.0277 (−30%) | 0.0307 (−23%) | 0.0969 | 0.0749 (−23%) | 0.0593 (−39%) |

[1] using Formula (21) with (4) and (5) from [8]. [2] using Formula (25) with (5) from [8].

## 5. Conclusions

The present study is devoted to analyzing the conditions of particle incipient motion from a sedimentation layer formed by hemispheres of the same radius as the particle. The main force parameters were sufficient for determining that the conditions of motion initiation were obtained as a result of numerical simulation of the Herschel–Bulkley shear flow around a particle located as close as possible to a rough surface. Based on the data obtained, the following conclusions can be drawn:

- The drag force was on average 6% higher in the case of the Newtonian fluid compared to the smooth surface. The hemisphere underlay increased the shear rate, thereby reducing the viscosity of the non-Newtonian fluid and decreasing the $C_d$ compared to the smooth wall. As the non-Newtonian properties of the fluid increased, the drag

force over the rough surface became smaller than over the smooth surface. Thus, the drag force was on average 25% lower for the case $Bn = 10$ and $n = 0.6$.

- The lacuna of the hemispheres under the particle increased the lifting force by 30% on average for a Newtonian fluid compared to a smooth wall. In the case of a non-Newtonian fluid, this effect was reduced due to viscosity reduction, but only at extreme values of $n$ and $Bn$ did the lifting force over a rough surface become lower than over a smooth one.
- The rough surface reduced the rotational torque of forces for a Newtonian fluid by 10% on average, and when the non-Newtonian properties of the fluid were increased in the considered regimes, the reduction was up to 40%.
- The application of forces obtained for a smooth wall in a non-Newtonian fluid underestimated the critical stresses for incipient motion and particle detachment for a rough wall by an average of 10% and 12%, respectively, with a maximum deviation of 35% and 47%, respectively.
- Applying the available correlations of Zeng et al. [8] to the forces obtained for a smooth wall and Newtonian fluid to a rough wall gave a noticeable difference in determining the critical stresses on the wall.

Approximation formulas have been suggested to describe the coefficients of the drag force $C_D$, lift force $C_L$, and torque $C_M$ acting on the particle. The average errors in the values were 2.5%, 3%, and 3.2%, respectively.

**Author Contributions:** Conceptualization, A.S., Y.S., and O.B.B.; methodology, A.S., Y.S., and O.B.B.; software, A.S. and Y.S.; validation, A.S.; formal analysis, A.S. and O.B.B.; investigation, A.S. and Y.S.; resources, O.B.B.; data curation, Y.I.; writing—original draft preparation, A.S.; writing—review and editing, O.B.B. and Y.I.; visualization, A.S.; supervision, O.B.B.; project administration, O.B.B.; funding acquisition, O.B.B. All authors have read and agreed to the published version of the manuscript.

**Funding:** This research was funded by Baker Hughes.

**Data Availability Statement:** The data presented in this study are available on request from the corresponding author.

**Acknowledgments:** The authors would like to thank the Baker Hughes company for permission to publish the research results.

**Conflicts of Interest:** The authors declare no conflict of interest. Yaroslav Ignatenko is employee of Baker Hughes. Alexander Seryakov is employee of OFS Technologies. Oleg Bocharov is employee of IWEP SB RAS. The paper reflects the views of the scientists, and not the companies or the institutions.

## Nomenclature

| | |
|---|---|
| $\dot{\gamma}$ | The second invariant of the strain rate tensor |
| $\delta$ | Vertical distance between the particle and the rough surface |
| $\theta$ | Ratio of the free-stream velocity over the rough surface to the smooth surface |
| $\mu$ | Fluid viscosity |
| $\rho, \rho_p$ | Density of fluid and particle |
| $\tau_y$ | Yield stress of fluid |
| $\tau_B$ | Shields number |
| $\tau_w$ | Shear stress on the wall |
| $\tau_L$ | Critical Shields number to detach particle |
| $\tau_R$ | Critical Shields number to roll particle |
| $A, B, C, D$ | Coefficients of approximation formulas |
| $a_{0...9}, b_{0...9}, c_{0...9}, d_{0...9}$ | Coefficients of approximation formulas |
| $C_D, C_L, C_M$ | Dimensionless coefficients of the drag and the lift force and the torque |
| $F_D, F_L, F_S$ | The drag force, lift force, and characteristic force |
| $G$ | Shear rate of the incoming free stream |
| $g$ | The gravity constant |
| $h$ | Distance of sphere to the x–y plane (flat surface) |
| $h_0$ | Minimum distance of the sphere to the x–y plane in case of rough surface |

| | |
|---|---|
| $k$ | Consistency factor |
| $m$ | Parameter for Papanastasiou regularization |
| $n$ | Power law index |
| $Re$ | Reynolds number |
| $Re_{SH}$ | Shear Reynolds number |
| $\boldsymbol{S}$ | Strain rate tensor |
| $\boldsymbol{u}$ | Fluid velocity vector |
| $u_p$ | Free-stream velocity in the front of the particle |
| $u_p^{smooth}$ | Smooth wall free-stream velocity in the front of the particle |

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
