# Peer review of "Characteristics of a Particle’s Incipient Motion from a Rough Wall in Shear Flow of Herschel–Bulkley Fluid"

_fluids, doi:10.3390/fluids9030065_

Round 1

Reviewer 1 Report

Comments and Suggestions for Authors

I would conditionally accept the article and recommend major changes to the current MDPI Fluids manuscript #280762. However, to increase the archival value of the article it would be prudent for the authors to respond to a few questions and make some corrections.

The authors used OpenFOAM CFD package utilizing SIMPLE-C algorithm to compute near rough wall shear-flow incipient motion of spherical particles in a laminar low-Re flow of rheological fluid (Herschel-Bulkley model). The authors also used a modelling approach to hydrodynamic coefficients. This is a complex article with many details, and it took this reviewer a lot of time to understand and review it.

Literature cited is decent and appropriate. Basic phenomena are well explained. Hydrodynamic lift, drag and pitching moment from a similarity law have been used to compute such forces and torques on spherical particles. Criteria and critical velocities have been discussed of incipient motion of spherical particles.

The authors put much effort into their work. Analytical and numerical methods have been used. Have the results been checked against the published experimental studies? I don’t think I saw that.

English is very good with few typos and omissions.

It would be appropriate if the author could provide some answers to questions. Here are also few general remarks:

  1. A nomenclature would be really very helpful. There are many definitions and summarizing them in one place helps.
  2. Can the authors in a sentence or two just clearly state what is new here that was not known or investigated before.
  3. Give some physical explanations of the observed phenomena. Authors claim that the pitching torque coefficients is larger for smooth walls than for rough walls. What would be the physical explanation for that (maybe I missed it).
  4. In reality due to side walls and particle collisions, yawing and rolling torques would exist. Any insight?
  5. Can the authors explain the relative amplitudes of hydrostatic (buoyancy) and hydrodynamic lift. It seems that buoyancy is a very significant part of the lift. Did perhaps coefficients of lift CL you use contain also hydrostatic lifting force. That would not be correct.
  6. Did the authors experience and if yes how did they resolve the issue of numerical instability in the vicinity of yield stress in H-B fluid?
  7. I suppose in your idealized treatment of 2D (horizontal- vertical) spherical particles only rolled (pitched over) along the longitudinal axis, right? No roll to the side or yaw?
  8. What effect does “Magnus” lift has on the total lifting force (Archimedean, Hydrodynamic lift, particle mechanical rotation)?
  9. Is there any slip (sliding) of spherical particles over the smooth or rough surface?
  10. Why would hydrodynamic drag on a particle increase in the case of the rough surface (Line 72)? Is it because of the velocity distribution in the boundary-layer or because the particle is sliding over the rough surface?
  11. What kind of computing power was used for the simulation? How long did the computations last? Any detail on required computational power and times is also helpful.
  12. The authors are mentioning for example in Eq. (5) the errors for the CL. How are these errors established? Based on what? Experiments?
  13. I guess the rolling motion really depends on the arrangement of hemispherical wall or relationship to fluid flow.

Some of the particular corrections needed in the article are:

1.     Line 0: In your title it should say “from” and not “From”.

2.     Line 36: Should say … Reynolds number and …

3.     Line 67: Herschel-Bulkley and not Herschel-Balkley.

4.     Line 167: It is more common to say “no-slip” condition or zero-slip rather than non-slip.

5.     Line 254: Lifting force is hydrostatic and hydrodynamic (and maybe Magnus) lift only. CL is just a coefficient of hydrodynamic lift apparently here.

6.     Line 257: Author talks about the CL in Eq. 5 and states what “v_s” is. Where is V_s in Eq. 5. CL’s here is nondimensional and functions of Reynolds number only. What Re number are the authors referring to?

7.     Line 277: Streamlines (one word).

8.     Line 320: You mean function B in q. 11 is evaluated according to Eq. 8 (not 9 as you state. That is for C).

9.     Line 335: put 4/3 in parenthesis such as (4/3)*PI*…

10.  Line 336: Archimedean force is also known as buoyancy or hydrostatic lift.

11.  Line 369: … have same behavior.

Comments on the Quality of English Language

English is overall very good with few typos and omissions. This reviewer has detected few typos or grammatic issues that need to be corrected. List is given. Few sentences can be polished.

Author Response

The authors are very grateful to the reviewer for his/her efforts and time in reviewing the article.

Overview

I would conditionally accept the article and recommend major changes to the current MDPI Fluids manuscript #280762. However, to increase the archival value of the article it would be prudent for the authors to respond to a few questions and make some corrections.

 The authors used OpenFOAM CFD package utilizing SIMPLE-C algorithm to compute near rough wall shear-flow incipient motion of spherical particles in a laminar low-Re flow of rheological fluid (Herschel-Bulkley model). The authors also used a modelling approach to hydrodynamic coefficients. This is a complex article with many details, and it took this reviewer a lot of time to understand and review it.

 Literature cited is decent and appropriate. Basic phenomena are well explained. Hydrodynamic lift, drag and pitching moment from a similarity law have been used to compute such forces and torques on spherical particles. Criteria and critical velocities have been discussed of incipient motion of spherical particles.

 English is very good with few typos and omissions.

It would be appropriate if the author could provide some answers to questions. Here are also few general remarks:

1.      Comment/Question/Remark

 The authors put much effort into their work. Analytical and numerical methods have been used. Have the results been checked against the published experimental studies? I don’t think I saw that.

            Answer

The correctness of the computations was verified through the comparison of the Newtonian shear flow calculations results with the data presented in (Lee,2017). In work (Lee, 2017), the authors validated the choice of the number of hemispheres for the formation of steady flow over a rough surface. Also, their results and computational technique parameters were verified against the known experimental results.

Here we simulate laminar isothermal steady-state flow. There is no models that should be validated.

2.      Comment/Question/Remark

 A nomenclature would be really very helpful. There are many definitions and summarizing them in one place helps.

            Answer

The Nomenclature has been added. See page 18.

3.      Comment/Question/Remark

Can the authors in a sentence or two just clearly state what is new here that was not known or investigated before.

Answer

The most significant conclusions have been added into the abstract.

4.      Comment/Question/Remark

Give some physical explanations of the observed phenomena. Authors claim that the pitching torque coefficients is larger for smooth walls than for rough walls. What would be the physical explanation for that (maybe I missed it).

Answer

The effect of the rough wall is opposite (Figure 7 b). It’s decreases torque acting on the particle. Physical explanation is added into the text (paragraphs that starts from sting 339).

The fluid that flows over the particle creates a positive torque acting on the particle. Correspondingly, the fluid flowing under the particle creates a negative torque acting on the particle. In the case of a rough surface, the lacuna under the particle reduces the total torque compared to a smooth surface because more fluid flow is able to flow under the particle.

5.      Comment/Question/Remark

In reality due to side walls and particle collisions, yawing and rolling torques would exist. Any insight?

Answer

Undoubtedly, additional forces and torque must arise due to interaction with other particles, and they may be important for particle motion, but here we are limited by the stationary laminar hydrodynamic formulation of the problem without interaction with other particles. Consideration of such effects is beyond the scope of the current formulation of the problem.

6.      Comment/Question/Remark

Can the authors explain the relative amplitudes of hydrostatic (buoyancy) and hydrodynamic lift. It seems that buoyancy is a very significant part of the lift. Did perhaps coefficients of lift CL you use contain also hydrostatic lifting force. That would not be correct.

Answer

Considered equations for steady-state flow does not include gravity forces, thus all obtained forces have no effect of buoyancy and depend on flow and fluid parameters. While solving mechanical problem of incipient motion and lift off the particle we add buoyancy force.

7.      Comment/Question/Remark

Did the authors experience and if yes how did they resolve the issue of numerical instability in the vicinity of yield stress in H-B fluid?

Answer

Numerical instability with infinite apparent fluid viscosity at low shear rate is resolved by Papanastasiou regularization. See paragraph starts with string number 161.

8.      Comment/Question/Remark

I suppose in your idealized treatment of 2D (horizontal- vertical) spherical particles only rolled (pitched over) along the longitudinal axis, right? No roll to the side or yaw?

Answer

Yes, correct. Geometry is fixed at all. Particle rolling is considered as static mechanical problem.

9.      Comment/Question/Remark

What effect does “Magnus” lift has on the total lifting force (Archimedean, Hydrodynamic lift, particle mechanical rotation)?

Answer

Moving or rolling of the particle was not considered while simulation.

10. Comment/Question/Remark

Is there any slip (sliding) of spherical particles over the smooth or rough surface?

Answer

Sliding is not considered in the current problem statement.

11. Comment/Question/Remark

Why would hydrodynamic drag on a particle increase in the case of the rough surface (Line 72)? Is it because of the velocity distribution in the boundary-layer or because the particle is sliding over the rough surface?

Answer

The drag force increases in case of Newtonian flow due to velocity redistribution. Lee and Balachandar (2017) were not considered particle motion while making this conclusion.

12. Comment/Question/Remark

What kind of computing power was used for the simulation? How long did the computations last? Any detail on required computational power and times is also helpful.

Answer

The modelling was performed using a corporate computing cluster. Some times after the work was completed, the cluster was rebuilt and some (or all) of the used computational nodes were upgraded (replaced). Unfortunately, at the moment it is not possible to find the technical parameters of the equipment used. Without it, computational time is useless.

13. Comment/Question/Remark

The authors are mentioning for example in Eq. (5) the errors for the CL. How are these errors established? Based on what? Experiments?

Answer

Eq. (5) was developed to calculate the CL at smooth wall. It was applied to calculation of CL at rough wall and compared with results of the current simulations to estimate error. Probably ‘error’ is not best word here. It will be replaced by ‘difference’ or ‘deviation’ to avoid misunderstanding.

14. Comment/Question/Remark

I guess the rolling motion really depends on the arrangement of hemispherical wall or relationship to fluid flow.

Answer

Yes, you're right. And also, the starting criterion depends on the slope of the underlying settled particles. Analyses of the criterion of particle incipient motion and particle detachment were carried out on the example of horizontal flow as an example to show the influence of rheology.

15. Comment/Question/Remark

Some of the particular corrections needed in the article are:

+1.     Line 0: In your title it should say “from” and not “From”.

+2.     Line 36: Should say … Reynolds number and …

+3.     Line 67: Herschel-Bulkley and not Herschel-Balkley.

+4.     Line 167: It is more common to say “no-slip” condition or zero-slip rather than non-slip.

+5.     Line 254: Lifting force is hydrostatic and hydrodynamic (and maybe Magnus) lift only. CL is just a coefficient of hydrodynamic lift apparently here.

Replaced by ‘lift force coefficient’ if I understood reviewer correctly.

+6.     Line 257: Author talks about the CL in Eq. 5 and states what “v_s” is. Where is V_s in Eq. 5. CL’s here is nondimensional and functions of Reynolds number only. What Re number are the authors referring to? There was typo.

Re is determined by the flow velocity at the center of the particle.

+7.     Line 277: Streamlines (one word).

+8.     Line 320: You mean function B in q. 11 is evaluated according to Eq. 8 (not 9 as you state. That is for C).

Everything is correct. Same polynomial function but with different coefficients. The same as for the lift force coefficient.

+9.     Line 335: put 4/3 in parenthesis such as (4/3)*PI*…

+10.  Line 336: Archimedean force is also known as buoyancy or hydrostatic lift.

+11.  Line 369: … have same behavior.

Reviewer 2 Report

Comments and Suggestions for Authors

The work needs substantial improvements. The following major points must be addressed carefully by revising the manuscript:

1.      The title must be modified.

2.      The paper must be entirely revised with the help of an English native narrator. The present form is not at the level of quality expected by the journal.

3.      Abstract should be rewritten highlighting the novel quantitative findings/ outcome of the study.

4.      The novelty of the work needs to be more explicitly stated through extensive literature reviews. There are significant volumes of work on the above topic. Check about the contribution to the current state-of-art in the considered area. 

5.      Author should list out all the basic assumption for the numerical solution.

6.      The mathematical treatment part is also not lucid. Equations are given, but what is physics behind the individual terms is not discussed in detail.

7.      Computational aspects/ technique/ should be explained adequately. It is highly recommended to add an algorithm of solution showing which equation and in which order they are being solved.

8.      For computations of the present model, following queries are to be answered:

a)      Grid study,

b)      Schematics of Mesh,

c)      Mesh description,

d)     CPU run time,

e)      Residual error

f)       Thermal performance evaluation criteria must be studied

9.      The model validation study is incomplete and not a good practice to present a comparison. The validation study should be rigorous (!) and detailed (!) with experiments, at least for a related problem. In fact, there is no validation study for the entropy generation.

10.  It would be appreciated if the authors include the numerical uncertainty analysis in the study.

11.  Physical explanation on the results and discussion section needs substantial improvements.

12. The conclusion should be re-written highlighting the novel quantitative findings alone.

13.  Based on the study, author should recommend for the best geometric as well as flow controlling variables.

Comments on the Quality of English Language

Need improvements.

Author Response

The authors are very grateful to the reviewer for his/her efforts and time in reviewing the article.

Overview

The work needs substantial improvements. The following major points must be addressed carefully by revising the manuscript:

1.      Comment/Question/Remark

The title must be modified.

Answer

The authors do not fully agree with this statement. It is difficult to formulate briefly all the goals of the current work. From our point of view, the current title of the article reflects its content and is laconic.

2.      Comment/Question/Remark

The paper must be entirely revised with the help of an English native narrator. The present form is not at the level of quality expected by the journal.

Answer

The article has been revied with MDPI English Language Editing Service.

3.      Comment/Question/Remark

Abstract should be rewritten highlighting the novel quantitative findings/ outcome of the study.

Answer

The most significant conclusions have been added into the abstract.

4.      Comment/Question/Remark

The novelty of the work needs to be more explicitly stated through extensive literature reviews. There are significant volumes of work on the above topic. Check about the contribution to the current state-of-art in the considered area. 

Answer

The topic of multiphase flow and calculation forces acting on particles is very wide, but topic of this study is very narrow. The goal of the article is investigation of forces and torque acting on a sphere on a bedload in shear laminar flow on Herschel-Bulkley fluid. This task is solved for Newtonian fluid in laminar case by Lee, 2017. Their results and computational technique parameters were verified against the known experimental results. Here we continue study for Herschel-Bulkley fluid. We think that review of all appropriate studies are already included in literature review of the article. An unreasonable increase in the number of articles in the introduction will take the reader away from the main objective of this paper.

The novelty of the paper is restated more explicitly at the end of the introduction.

5.      Comment/Question/Remark

Author should list out all the basic assumption for the numerical solution.

Answer

Laminar steady-state flow is assumed. Boundary condition is described. Simulation is made using well known SIMPLE-C algorithm realized in OpenFOAM in simpleFoam solver. This information is available in sections «Problem statement” and “Numerical algorithm”.

6.      Comment/Question/Remark

The mathematical treatment part is also not lucid. Equations are given, but what is physics behind the individual terms is not discussed in detail.

Answer

Navier-Stokes and continuity equation are the basic equations of fluid mechanics. We don’t see necessity to discuss them.

7.      Comment/Question/Remark

Computational aspects/ technique/ should be explained adequately. It is highly recommended to add an algorithm of solution showing which equation and in which order they are being solved.

Answer

Description of numerical algorithm and numerical schemes was extended. SIMPLE-C is well known CFD algorithm for solving Navier-Stokes equations. We don’t see necessity to describe it the article.

8.      Comment/Question/Remark

For computations of the present model, following queries are to be answered:

  1. Grid study – table and paragraphs were added
  2. Schematics of Mesh – visualization of mesh was added
  3. Mesh description – it was mentioned in original article; mesh description was modified
  4. CPU run time – average simulation time was added
  5. Residual error – it was mentioned in original article, 10-9
  6. Thermal performance evaluation criteria must be studied – problem statement is isothermal; such tasks are lying out of goals of the current article

9.      Comment/Question/Remark

The model validation study is incomplete and not a good practice to present a comparison. The validation study should be rigorous (!) and detailed (!) with experiments, at least for a related problem. In fact, there is no validation study for the entropy generation.

Answer

The correctness of the computations was verified through the comparison of the Newtonian shear flow calculations results with the data presented in (Lee,2017). In work (Lee, 2017), the authors validated the choice of the number of hemispheres for the formation of steady flow over a rough surface.

Here we simulate laminar isothermal steady-state flow. There are no models that should be validated.

The problem formulation is isothermal, consideration of entropy generation is beyond the scope of the problem.

10. Comment/Question/Remark

It would be appreciated if the authors include the numerical uncertainty analysis in the study.

Answer

It is not possible to quantitatively analyze the numerical error because the grid is unstructured. This information is added to the paragraph discussing mesh refinement.

11. Comment/Question/Remark

Physical explanation on the results and discussion section needs substantial improvements.

Answer

A qualitative physical explanation of the behavior of the drag force, lift force and force-moment curves is given in the text. We see no way of explaining the behavior of forces and moments in more detail.

12. Comment/Question/Remark

The conclusion should be re-written highlighting the novel quantitative findings alone.

Answer

In our opinion, the concluding section summarizes all the conclusions worthy of note.

13. Comment/Question/Remark

Based on the study, author should recommend for the best geometric as well as flow controlling variables.

Answer

The article provides approximating formulae for describing forces and torque that can be applied in practice. The critical Shields parameters are derived to help the engineer to estimate the possibility of sediment movement or he may use other approach at his own discretion that includes for example force coefficients.

Reviewer 3 Report

Comments and Suggestions for Authors

This paper perform a numerical simulation of Herschel-Bulkley shear flow around a stationary particle located on sedimentation layer. The authors then studied the particle lift force over roughness, and also analyze the effect of Bingham number. The authors also propose several formula for coefficients of drag force, lift force and torque acting on the particles. This is paper is well written and interesting. Some minor suggestions: 

The notation of 'ReSH = 2 ÷ 200, n = 0.6 ÷ 1.0 and Bn = 0 ÷ 10' is very confusing and the authors do not define these notation, either in the abstract or in the main text. Are they just standard division? If so, perhaps just simplify the number into 'ReSH = 0.01, n = 0.6 and Bn = 0'. I am not clear why the authors want to write the numbers in this way. 

A paper organization is suggested to be added in the end of the introduction. 

Table 2, Zeng seems to a reference and it is suggested to add reference number there. 

Figure 6, it is suggested also clarify the meaning of solid lines. 

Did the authors use any turbulence model or this paper is only for laminar steady-state solution?

Right after equations like (21) and (22), there does not need a space before 'where ...'. 

Author Response

The authors are very grateful to the reviewer for his/her efforts and time in reviewing the article.

Overview

This paper perform a numerical simulation of Herschel-Bulkley shear flow around a stationary particle located on sedimentation layer. The authors then studied the particle lift force over roughness, and also analyze the effect of Bingham number. The authors also propose several formula for coefficients of drag force, lift force and torque acting on the particles. This is paper is well written and interesting. Some minor suggestions: 

1.      Comment/Question/Remark

 The notation of 'ReSH = 2 ÷ 200, n = 0.6 ÷ 1.0 and Bn = 0 ÷ 10' is very confusing and the authors do not define these notation, either in the abstract or in the main text. Are they just standard division? If so, perhaps just simplify the number into 'ReSH = 0.01, n = 0.6 and Bn = 0'. I am not clear why the authors want to write the numbers in this way.

Answer

In Russian, Poland and Italian books sign ”÷” can mean range. It removed to avoid misunderstanding.

2.      Comment/Question/Remark

 A paper organization is suggested to be added in the end of the introduction.

Answer

The paragraph with article organization is added in the end of introduction.

3.      Comment/Question/Remark

 Table 2, Zeng seems to a reference and it is suggested to add reference number there. 

Answer

The appropriate reference has been added.

4.      Comment/Question/Remark

 Figure 6, it is suggested also clarify the meaning of solid lines.

Answer

Appropriate information has been added to the figure description.

5.      Comment/Question/Remark

 Did the authors use any turbulence model or this paper is only for laminar steady-state solution?

Answer

Laminar steady-state solution only. Appropriate changes have been made in abstract, problem statement.

6.      Comment/Question/Remark

 Right after equations like (21) and (22), there does not need a space before 'where ...'. 

Answer

Multiple correction of such typos in the text was made.

Round 2

Reviewer 1 Report

Comments and Suggestions for Authors

Accept in current form with minor English editing - final check.

Comments on the Quality of English Language

Minor English editing - final check.

Reviewer 2 Report

Comments and Suggestions for Authors

The Manuscript has been improved substantially in this revision.